# Cervical length varies considering different populations and gestational outcomes: Results from a systematic review and meta-analysis

T. G. Bortoletto[1], T. V. Silva[1,2], A. Borovac-Pinheiro[1], C. M. Pereira[1], A. D. Silva[1], M. S. França[3], A. R. Hatanaka[3], J. P. Argenton[1], R. Passini, Jr.[1], B. W. Mol[4], J. G. Cecatti[1], R. C. Pacagnella[1]*

1 School of Medical Sciences, University of Campinas (UNICAMP), Campinas, São Paulo, Brazil,
2 University of Pernambuco (UPE), Recife, Pernambuco, Brazil, 3 Department of Obstetrics and Gynaecology, Federal University of São Paulo (UNIFESP), São Paulo, Brazil, 4 Department of Obstetrics and Gynaecology, Monash University, Clayton, Victoria, Australia

* rodolfop@unicamp.br

**Data Availability Statement:** All relevant data are within the manuscript and its Supporting information files.

## Abstract

### Background

The uterine cervical length is an important risk factor for preterm birth. The aim of this study was to assess cervical length distribution in women with singleton pregnancies, measured by transvaginal ultrasound between 16 and 24 weeks, and its association with population characteristics.

### Materials and methods

We searched electronic databases and other sources for studies published from April 1, 1990 to July 21, 2020. Of the 2019 retrieved publications, full-text versions of 137 articles were considered. We included 77 original articles that reported cervical length measurements of 363,431 women. The main aim of this study was to identify the pattern of cervical length in different populations. We collected demographic and clinical data concerning the population, in addition to information regarding the ultrasound examination and cervical length measurement. Regarding study bias, 56 were at low risk of bias and 21 were at medium risk of bias.

### Results

The meta-analysis included 57 articles with data from 158,346 women. The mean cervical length was 37.96. mm (95% CI [36.68, 39.24]). Cervical length was shorter in women from Africa and Asia, in those from low-income countries, with a lower body weight, and in those who delivered before 37 gestational weeks. We found that the cervical length from pooled studies is longer than that usually discussed in the literature. Regarding limitations, we had difficulty assessing our main variable because there was no consistent pattern in the way

**Funding:** This work was supported by the Bill & Melinda Gates Foundation, Seattle, WA [OPP1107597], the Brazilian Ministry of Health and the Brazilian National Council for Scientific and Technological Development (CNPq) [401615/20138]. RCP and BM received funds for the study. No author received salary from any funders and the funders had no role in study design, data collection and analysis, decision to publish, or preparation of the manuscript.

**Competing interests:** The authors have declared that no competing interests exist.

authors reported cervical length measurement. Another limitation was the great heterogeneity between studies.

## Conclusions

The use of a single cutoff value to define a short cervix diagnosis, an important risk factor for preterm birth, may not be correct and cervical length must be considered according to maternal population characteristics. Future studies should identify different specific curves and cutoff values for cervical length in different populations. This meta-analysis was registered in the PROSPERO database under CRD42017070246 at https://www.crd.york.ac.uk/prospero/display_record.php?RecordID=70246.

## Introduction

Around 15 million preterm births occur every year worldwide [1]. Prematurity is the primary cause of neonatal death and morbidity around the globe [2], and the earlier the gestational age (GA) at birth, the greater the associated risks [2]. Aside from multiple pregnancy [3] and obstetrical history of a previous preterm birth [4], a short uterine cervix has also been found to be associated with premature delivery, and its presence can be evaluated by transvaginal ultrasound during the second trimester, which allows for risk assessment [5–8].

The transvaginal ultrasound (TVUS) technique is well established [9]. The reference ranges for distribution curve and cervical length percentiles were first defined in the 90s. Since then, several researchers have used these cutoff limits as standards. Iams et al. [10] found the smallest risk of delivering before 35 weeks in women whose cervical length was over 40 mm (75th percentile) and established all comparisons based on this cutoff.

Major trials chose lower limits to propose interventions [5, 11–14], probably also considering the values offered by Iams et al. [10] and other researchers [15–18], in which cutoffs like 15, 20, 25 or 30 mm were suggested, although the most agreed upon value is currently 25 mm. Official guidelines do not recommend transvaginal ultrasound as part of a universal screening program [19–21], especially when women have no history of spontaneous preterm birth. However, they recognize its value as long as clinicians remain aware of the real indications in prescribing (or not) specific interventions [22, 23]. The confidence to do so relies on determining the normality according to the gestational week of screening and maternal characteristics.

In order to understand the distribution of cervical length and its classification as normal or abnormal, this review systematically evaluated original research that reported transvaginal ultrasound imaging of cervical length measurements in women with singleton pregnancies between 16 to 24 gestational weeks. The aim of this study was to identify the cervical length distribution in different populations to guide clinical practice considering population characteristics. We decided to evaluate the cervical length between 16 to 24 gestational weeks because this is the time at which the largest number of interventions to prevent preterm birth are proposed.

## Materials and methods

This systematic review and meta-analysis was developed according to the Preferred Reporting Items for Systematic Reviews and Meta-Analysis (PRISMA) statement [24] (S1 Appendix) and registered under the identification number CRD42017070246 in the PROSPERO database. All

articles were assessed by two independent researchers who retrieved and reviewed studies for eligibility, assessed their risk of bias and extracted data. Divergences were resolved by a senior researcher.

We searched Medline, Embase, Scielo and clinicaltrials.gov, as well as the references of retrieved articles, to identify original papers that performed transvaginal ultrasound with rigorous imaging criteria from 16 to 24 gestational weeks in women with singleton pregnancies to evaluate cervical length in different populations. We only included studies that provided the prevalence of different cervical length measurements with an updated technique; therefore, we searched for randomized controlled trials, cohort and observational studies published between April 1, 1990 to July 21, 2020, with no language restriction. We used a combination of terms related to preterm birth and cervical length for the search strategy (see S2 Appendix for the complete search strategy). Inclusion criteria were as follows: population of women with singleton pregnancies universally evaluated from 16 to 24 gestational weeks; detailed description of TVUS; and published from April 1, 1990 to July 21, 2020. We only included studies that had described the cervical measurement technique adequately and studies that followed Fetal Medicine Foundation cervical assessment orientation.

The exclusion criteria were unclear, absent or outdated TVUS technique description; cervical length measurement after threatened preterm labor; women with symptoms; or those already submitted to cervical cerclage in the current pregnancy.

Once eligible articles had been defined, we extracted information regarding the method (design, TVUS technique, sampling, statistical analysis); country of origin; number of subjects; gestational age at sonography; risk of prematurity (history of preterm birth or cervical procedures, as established by the original authors); cervical length; gestational age at birth; and obstetric, demographic and anthropometric maternal characteristics. In cases where the original article described a gestational age surpassing the scope of the review, we only analyzed these data if the authors made cervical length measurements between 16 and 24 gestational weeks. All included articles collected information concerning the risk of prematurity; however, the classification of women as high or low risk was not homogeneous across the studies (described in S3 Appendix).

After analysis of the full text, excluded articles were arranged in a table according to reason for exclusion, and the included articles were subject to quality assessment using the National Heart, Lung and Blood Institute [25] tool (available in S4 Appendix).

The selected publications that provided cervical length measurements as a continuous variable (mean or median and their standard deviation or interquartile range) were analyzed using meta-analysis to generate a single mean value and its standard deviation. This process was applied to infer mean cervical length measurements for all women (and later, a different meta-analysis excluding those with previous conization), women split according to the economic development status of their continent of origin (low/middle- or high-income country) as defined in the World Development Indicators [26]; and as low or high risk of preterm delivery. It is important to note that the classification criteria used to define low and high risk for preterm birth by the authors were not clear or similar. Some characteristics such as previous preterm birth, cervical procedures or Mullerian malformation were cited for some authors, but overall, there was no complete description of the specific features to classify woman as low or high risk for prematurity. Comparisons estimated the difference in cervical length according to parity, age and birth outcome.

In papers that provided data as categorical variables, we reported the absolute number of individuals who presented in each cervical length stratum for statistical analysis. The Aleatory model was used to estimate the proportion of women in each cervical length stratum. In papers that reported continuous variables, we used meta-analysis to calculate the mean and standard

deviation for cervical length measurement and the proportion of individuals in each cervical length stratum. From publications that reported percentiles, we calculated the mean and standard deviation for each percentile and constructed a descriptive theoretical distribution curve for the entire period from 16 to 24 gestational weeks.

We also analyzed aggregated cervical length breakdown by continents, country income, women's BMI and pregnancy outcome (preterm or term delivery). We used R software, version 3.4, package metaphor 2.0–0 from the R Foundation for Statistical Computing for statistical analysis. The dataset is available as S1 Data.

RCP and BM received funds from the Bill & Melinda Gates Foundation, Seattle, WA [OPP1107597], the Brazilian Ministry of Health and the Brazilian National Council for Scientific and Technological Development (CNPq) [401615/20138] for the study. No author received salary from any funders and the funders had no role in study design, data collection and analysis, decision to publish, or preparation of the manuscript.

## Results

We identified 4089 articles through the databases and 23 from other sources, including reference search and experts' suggestions. After removing duplicates, we screened 2825 records, of which 162 were found to be eligible. We then excluded 85 articles due to the following reasons: method (one paper), no description of the screened women (10 papers), ineligible population (18 papers), TVUS technique different from established patterns or not reported (15 papers), cohort already included in another publication (13 papers), and gestational age surpassing the scope of the review (28 papers). Fig 1 shows the PRISMA flow diagram and S1 Table details all excluded articles and the reasons for exclusion.

The remaining 77 articles (64 cohorts, 10 clinical trials and 3 cross-sectional study) included 363,431 women. We included one study from South Africa, Belgium, Botswana, Catalonia, Croatia, Egypt, Finland, Greece, Hong Kong, India, Iran, Italy, Japan, Romania, South Korea, Sweden, Taiwan, two studies from Tukey and UK, three from The Netherlands and Thailand, four from China and Spain, ten from Brazil and 24 from the United States. Also, five studies included women from two or more countries.

The papers were evaluated according to the mode of describing the main variable, the cervical length. The four overlapping categories were: mean cervical length (57 articles), number of women in each stratum of cervical length (52 articles), percentile (15 articles), and mean cervical length according to gestational week (9 articles) (S2 Table).

In the quality assessment, 56 papers were determined to have a low risk of bias, 21 had medium risk, and none were considered to have a high risk of bias (S5 Appendix). Nine categories presented homogeneous performance, with over 62 studies complying with high quality criteria. Inferior performance was presented in two categories based on the description of statistical analysis (item 5 with 29 papers complying, and item 14 with 39 papers complying): multiple assessment of exposure (item 10 with 22 papers complying) and considerations about blinding of assessors (item 12 with 12 papers complying). Considering the variable of exposure (item 8), 39 articles provided both categorical and continuous variables, 19 articles presented continuous variables and 19 provided categorical variables.

The meta-analysis included 57 publications, accounting for 158,346 women. Considering the Q test of heterogeneity, studies were considered heterogeneous ($p < 0.0001$), with an $I^2$ of 100%. The mean cervical length was 37.96. mm (95% CI [36.68, 39.24]); Fig 2). Year of publication did not influence cervical length.

In a second different meta-analysis, we included only studies that studied a population without previous cervical conization (S1 Fig). This second meta-analysis included 40 publications

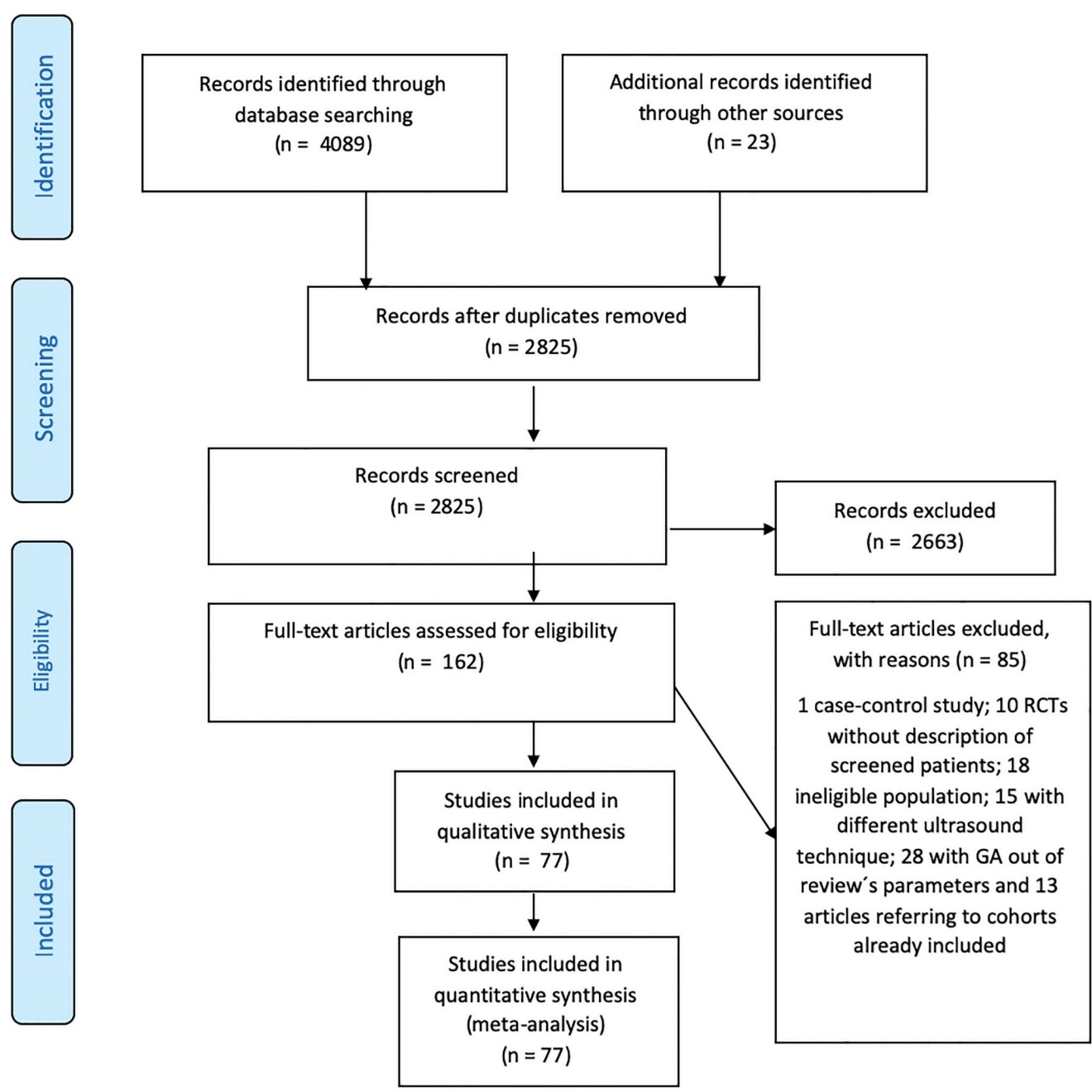

**Fig 1. PRISMA diagram.** Flow diagram of included articles according to PRISMA 2009 guidelines.

and a total of 91,408 women. The mean cervical length was 38.21 mm with a 95% CI of 36.70 to 39.71 mm.

When cervical length was evaluated according to the continent of origin, the mean cervical length was 32.77mm for Africa (95% CI [31.76, 33.78]), 36.59mm (95% CI [35.33, 37.84]) for South American, 39.37 mm (95% CI [35.34, 43.40]) for North American, 35.98mm (95% CI [34.21, 37.76]) for Asian and 39.81 mm (95% CI [38.23, 41.40]) for European populations (Fig 3).

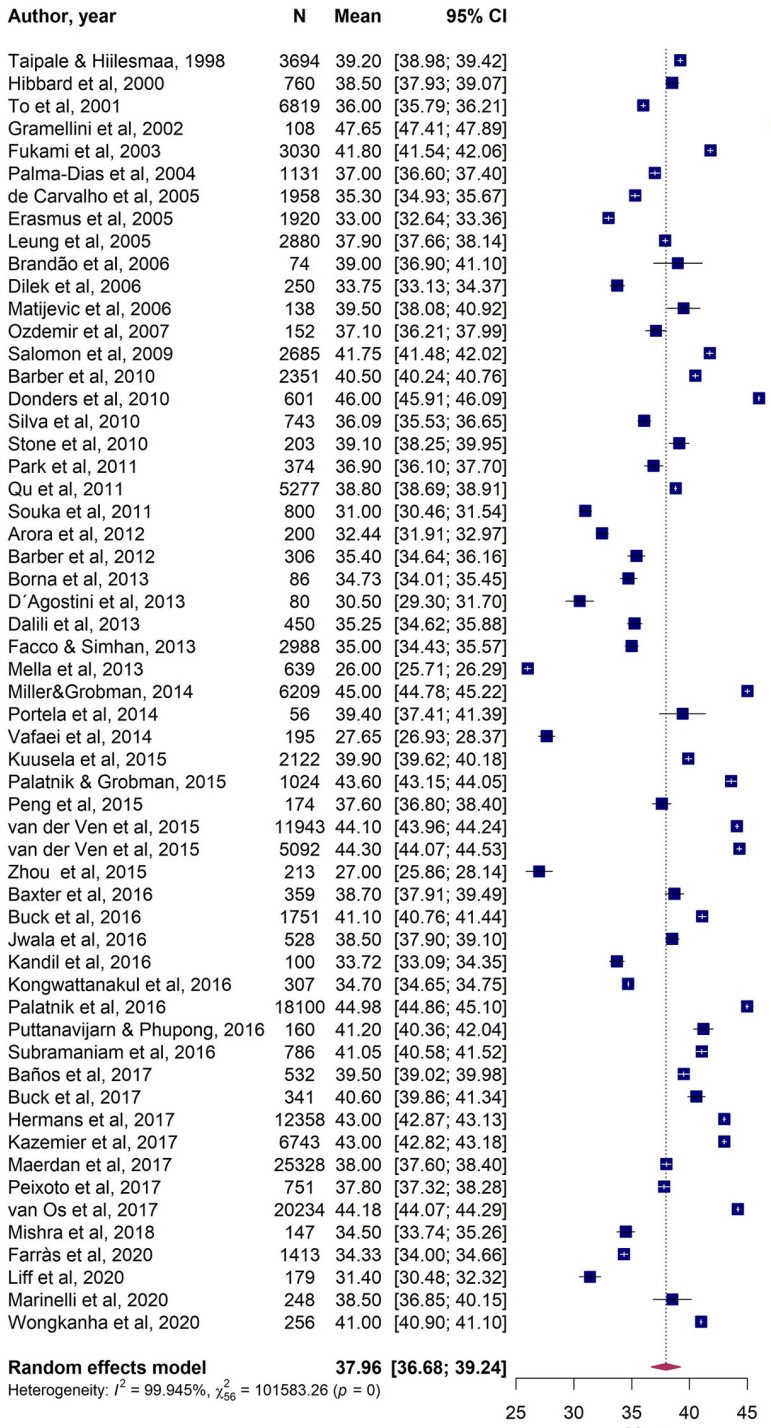

**Fig 2. Forest plot of cervical length measurements (in millimeters) of 57 publications included in the meta-analysis.**

Although the confidence intervals overlap with each other when analyzing data exclusively within a single continent, comparison of five continents using data from all articles included in the meta-analysis demonstrated a statistical difference between countries (p = 0). Fig 4 presents the difference between low/medium- and high-income countries, which presented

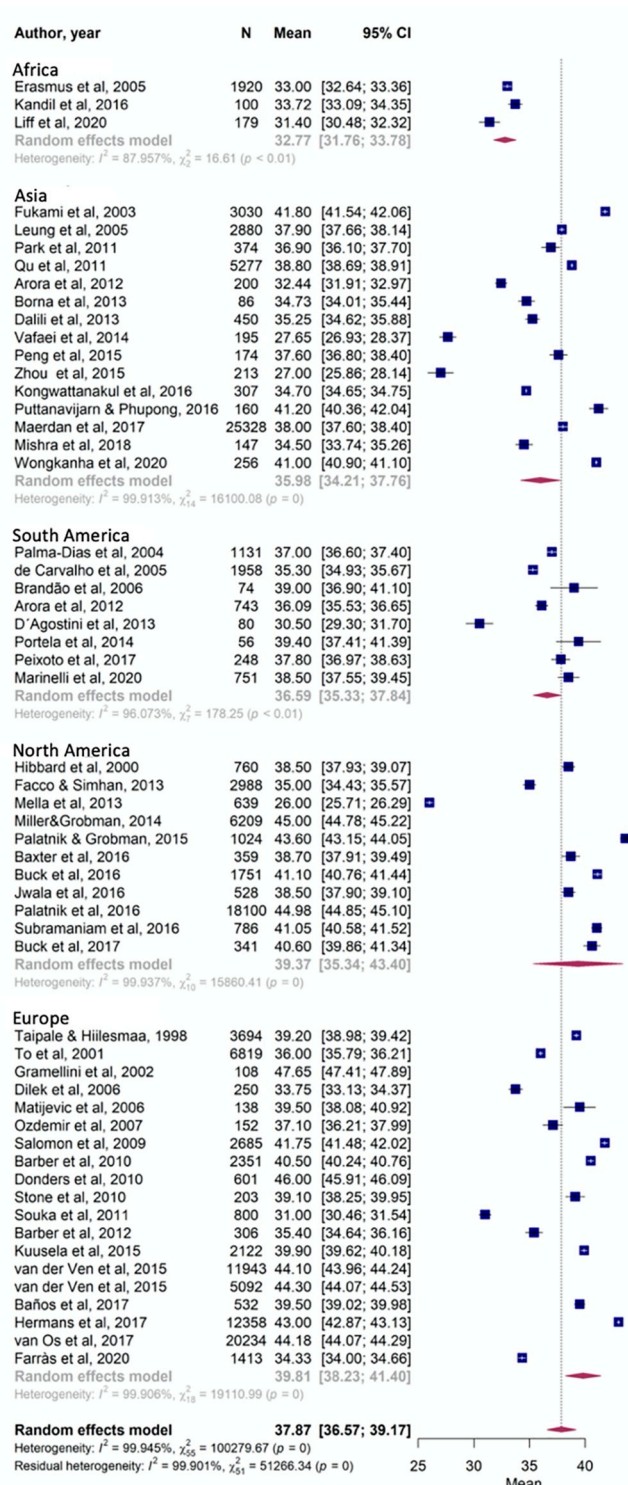

**Fig 3. Forest plots for the cervical length of women from each continent (Africa, South America, North America, Asia and Europe).**

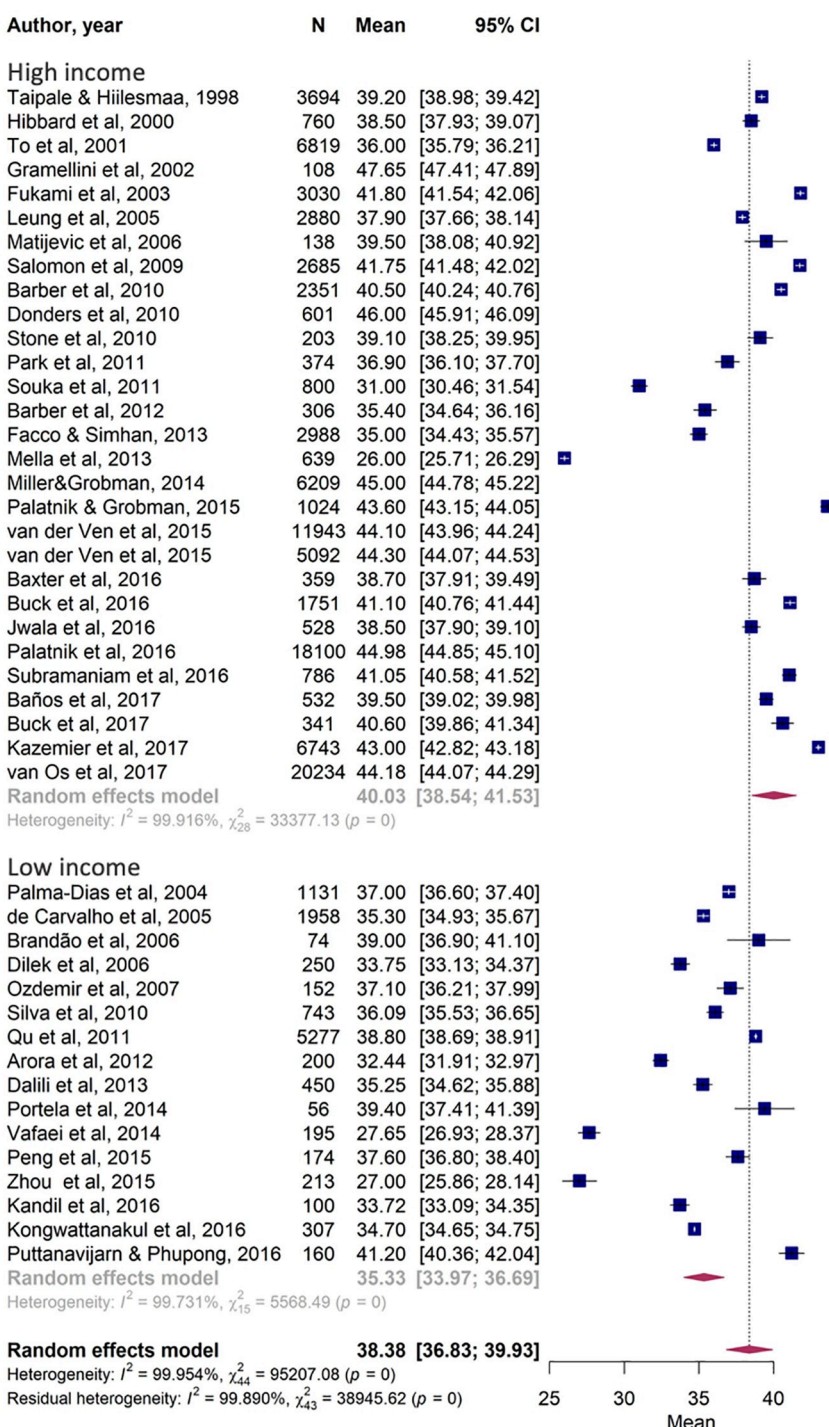

**Fig 4. Comparison of cervical length according to low/medium-income versus high-income countries.**

significantly different mean cervical lengths of 35.33 mm (95% CI [33.97, 36.69]) and 40.03 mm (95% CI [38.54, 41.53]), respectively (p = 0).

Cervical length was statistically shorter for nulliparous women; however, this finding was not clinically significant (mean difference 1.02 mm, 95% CI [-1.96, -0.07], S2 Fig). No

differences were found according to age (adolescents vs. adults; S3 Fig) or risk for preterm birth (low risk vs. high risk; S3 Appendix).

When comparing cervical measurements according to anthropometric features, we observed differences between women with the highest and lowest body mass index (BMI), with thinner women presenting cervical lengths 12.58 mm shorter than obese women (Fig 5).

Another comparison was regarding to birth outcomes. Ten publications (19,399 women) reported cervical length according to gestational age at birth. Preterm births (under 37 weeks) presented a mean cervical length -3.80 mm shorter than term births (95% CI [-5.15, -2.44], P<0.01; Fig 6).

Fifteen articles reported cervical length as percentiles, 14 of which used 26.8 mm as the cut-off for percentile 5, and the second most commonly reported one (by 13 articles) was percentile 50, corresponding to 38 mm. Fig 7 shows a descriptive theoretical distribution curve (red line) and compares it to those originally presented by Iams et al. (blue line).

When analyzing data according to cervical length cutoff values, 20 studies considered 20 mm as the cutoff limit, totaling 88,009 women, thus 3% of the study population were considered under the cutoff. For the 25 mm limit, there were 39 studies accounting for 146,500 women, 6,7% of whom were under the cutoff value. The 30 mm limit was used in 19 papers, including 88,380 subjects, with 13,1% of women classified as having a cervical length under the cutoff.

## Discussion

The main finding of this study is that cervical length ranges vary across populations when evaluated between 16 to 24 weeks gestation. Also, the cervical length was found to be shorter in women with a low body weight and in those who had a preterm birth.

Our largest meta-analysis of 57 publications included approximately one-third of the total number of women (158,346 out of 363,431) and reported a mean cervical length of 37.96 mm. This measurement was virtually the same (38.21 mm) when evaluating studies whose data did not include women who had undergone a previous cervical excisional procedure, which is consistent with the findings of other authors in publications describing cervical lengths, with no modifications in pregnancies observed after excisional procedures [27, 28].

Our main hypothesis was that cervical length would vary according to population characteristics. We demonstrated associations of cervical length with regional and demographic features when measurements were compared by continent and by income.

Mean measurements between continents presented clinically relevant differences of more than 6 mm when comparing Africa to North American or European women and almost 4 mm when comparing Asia to European women. Moreover, the statistical analysis presented a significant p-value. When countries were grouped according to income, which meant transferring data from Japan, Hong Kong and South Korea from the Asian group to the high-income group and transferring data from Turkey from the European group to the middle/low-income group, we also identified a clinically relevant and statistically significant difference over 4mm. High-income countries presented a mean cervical length of 40.03 mm while middle/low-income countries presented 35.33 mm.

When evaluating a population exclusively from the Netherlands, the ProTWIN Study reported mean cervical length measurements ranging from 43.6 to 44.2 mm, and the 25th percentile was 38 mm in 813 twin pregnancies from 16 to 22 weeks [29]. Although our review refers to single pregnancies only, we consider the ProTWIN Study as representative of high-income countries. This trial corroborates the idea of population differences in cervical length measurements.

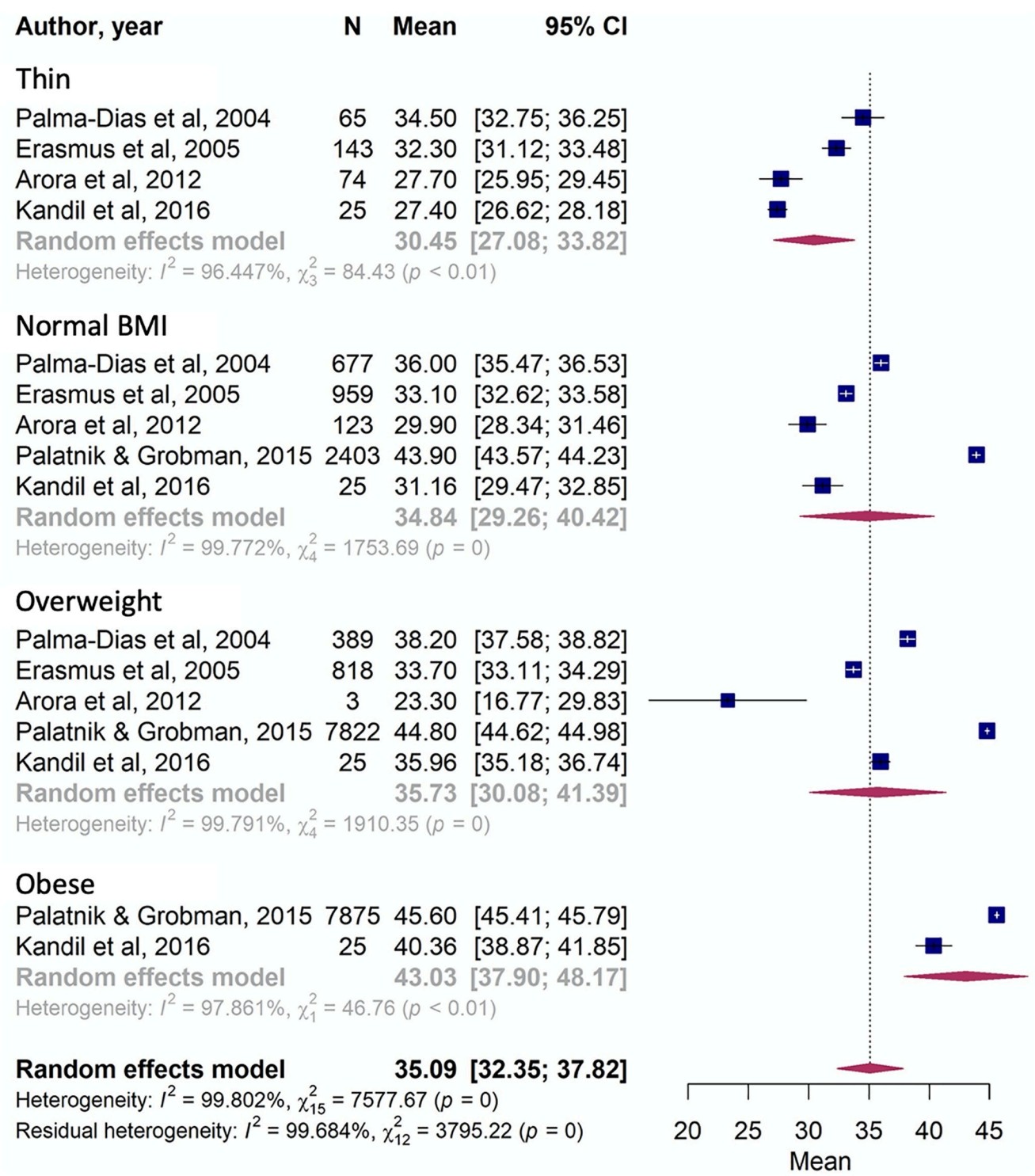

**Fig 5. Comparison of cervical length according body mass index (BMI).**

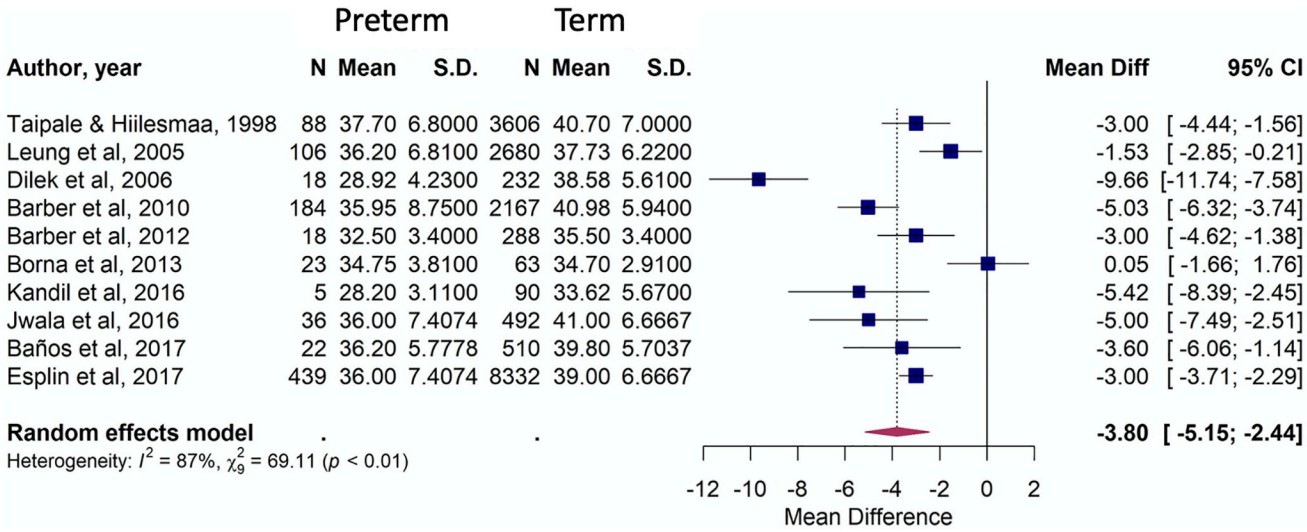

**Fig 6. Forest plots of mean cervical length measurements in pregnancies evolving to term and preterm births.**

It remains unclear whether the difference in cervical length between populations is due to environmental factors that would interfere in population growth capability, such as nutrition and access to health services, or due to intrinsic characteristics such as ethnicity/race [30] and anthropometric features. It is clear that there is an important association amid socioeconomic status and race in all western world, nevertheless, few included studies described information of short cervix for race/skin color groups and, in our study, it is difficult to analyze this data and infer association between race and short cervix without possible bias. However, it is an ecological approach and it is important to acknowledge that this association does not directly infer that shorter cervical lengths were more common in socially and economically vulnerable women. To clarify this question, future studies on cervical length should consider individual socioeconomic characteristics in the same population.

Socioeconomic characteristics are only starting points to consider when proposing limits of normality in studies and guidelines. A defined cutoff point represents a dichotomous boundary from which clinicians begin to consider different risk levels, and it is yet to be established if the 5 mm cervical shortening in women from low-income countries seen in our review is indeed a determining factor for preterm birth. Considering the same cutoff points for different population characteristics, perhaps doctors are deciding whether or not to intervene based on an incorrect level of association between the risk of preterm birth and the cervical length measurement.

Anthropometric characteristics also presented a significant difference when thin women were compared to women with obesity. The first group had a clinically relevant 12.58 mm shorter cervix. This result is consistent with The Preterm Prediction Study, where non-obese women had mean cervical measurements shorter (34.9 mm) than obese women (36.5 mm) [31]. However, our study identified an even greater difference between these two groups. Women with lower weight gain during pregnancy present a more significant odds ratio of spontaneous preterm birth [3]. In constrast, overweight and grade I obese women have a lower risk of spontaneous preterm delivery, and maternal BMI has a different effect according to different etiological subtypes of preterm birth [32]. This should be explored in further studies and considered in terms of a possible association with confounding factors such as income and maternal age.Another hypothesis of the current review was that the uterine cervix would

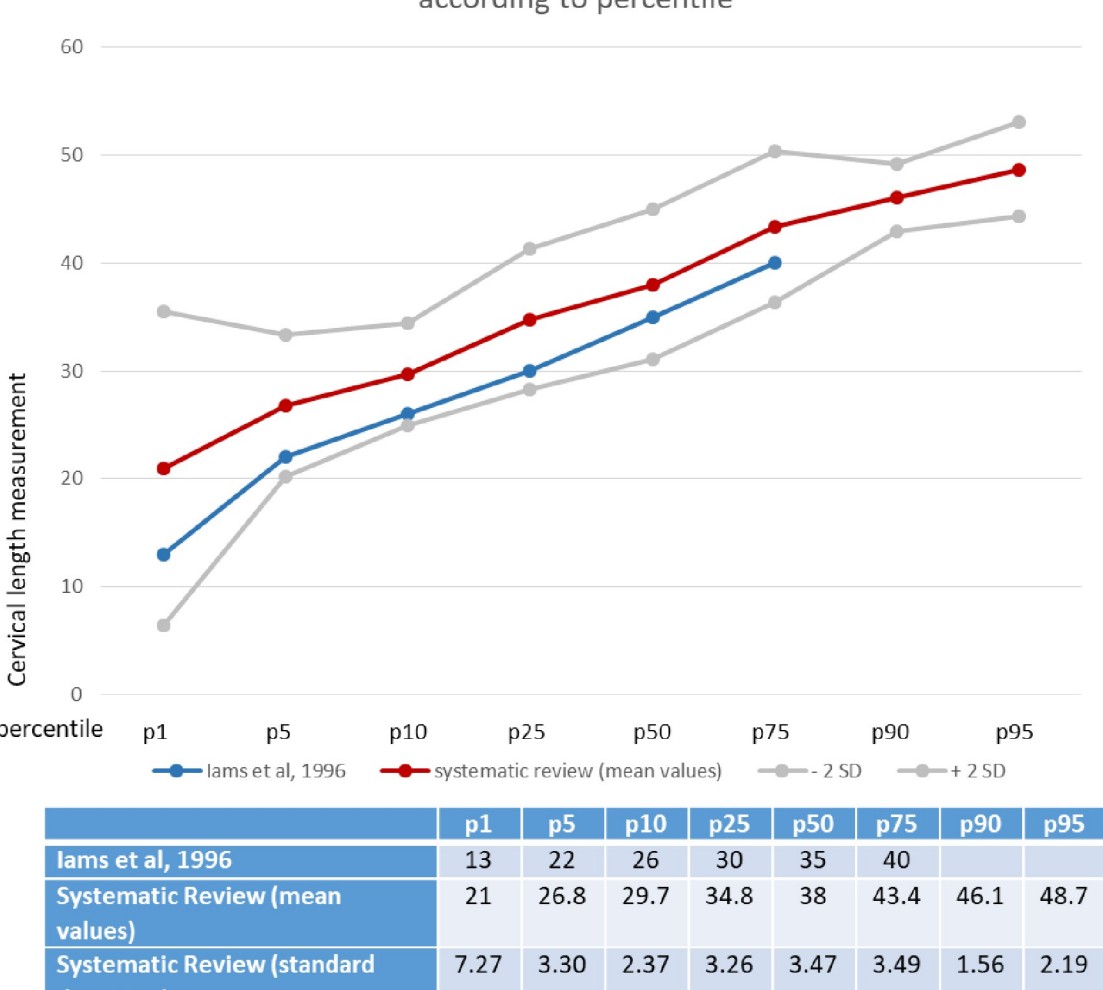

**Fig 7. Cervical length measurement according to percentile.**

| | p1 | p5 | p10 | p25 | p50 | p75 | p90 | p95 |
|---|---|---|---|---|---|---|---|---|
| Iams et al, 1996 | 13 | 22 | 26 | 30 | 35 | 40 | | |
| Systematic Review (mean values) | 21 | 26.8 | 29.7 | 34.8 | 38 | 43.4 | 46.1 | 48.7 |
| Systematic Review (standard deviation) | 7.27 | 3.30 | 2.37 | 3.26 | 3.47 | 3.49 | 1.56 | 2.19 |

be shorter in women who had a preterm birth. We retrieved 10 articles using the homogeneous reference of birth before 37 gestational weeks, and were able to separate measurements for groups of women who had a term or preterm birth. The results showed statistically significant and clinically relevant differences, with the preterm group found to have a mean cervical length measurement 3.80 mm (-5.15, -2.44) shorter than the term group. A retrospective cohort involving 17,295 women identified a higher rate of preterm birth in asymptomatic women with a short cervix when compared to the general cohort (40.4% versus 8.7%, $p < 0.001$) [33]. This corroborates the use of cervical length measurement as an important predictor of prematurity and highlights the need to separate specific groups as previous preterm birth and parity in the analysis to establish clear parameters of normality and interventions performing an IPD analysis.

The corresponding cutoff values observed for each percentile in the current study were similar to those defined by Iams et al. [10], with an increase of approximately 3 to 4 millimeters in

each corresponding percentile. Although we cannot present our data as an actual distribution curve, we can argue that these are theoretical values from which we can begin to delineate a population reference range.

Historically, Iams' 10th percentile (rounded to 25 mm) was chosen as a landmark for increased risk; however, it has been proposed that better predictive accuracy is achieved using thresholds of 20 mm or less, depending on the population studied [34]. Considering results from a cohort developed in our hospital, the comparable percentiles from 21 to 24 weeks were even longer, with a 10th percentile of 33.9 mm, 50th percentile of 41.4 mm and 90th percentile of 51.7 mm [35]. A cohort performed with nulliparous women in Switzerland demonstrated similar data, with mean cervical lengths of 40.3 mm at 20 weeks, 38.7 mm at 21 weeks, 39.6 mm at 23 weeks and 41.4 mm at 24 gestational weeks [36]. Furthermore, in a Japanese cohort, the mean cervical length at 16 weeks was 43 mm [37]. These differences could be explained by the population selection of exclusively low-risk women that evolved to term deliveries, but they could also suggest that the cutoff points commonly used in clinical practice may be underestimated.

Considering the limitations of our review, we had difficulty appraising our primary variable because there was no consistent pattern in the way authors reported cervical length measurement. Even if different publications were considered to be in the same category, sometimes the authors diverged concerning the cutoff limits chosen or the reported percentiles. Some authors reported percentiles as median and interquartile range [10, 38–41], while others opted to report only percentiles under the 10th percentile [17, 42, 43]. As acknowledged in other systematic reviews [7, 8, 44], differences in definitions of reference values limit the ability to make comparisons in reviews.

Publications reporting results as categorical variables referred to the number of women with a cervical length measurement under a particular cutoff value. We identified 20 publications that evaluated the cutoff limit of 20 mm for 88,009 women, 39 articles evaluating 25 mm for 146,500 patients, and 19 papers evaluating 30 mm for 88,380 women. The proportion of cervix measurements under each one of these limits were 3%, 6,7% and 13,1%, respectively.

This information is relevant when we consider the relative risk of 3.8 under 30 mm, 6.2 under 26 mm and 9.49 under 22 mm, as proposed in 1996 by Iams et al. [12]. This means that around 13% of the worldwide population have almost four times higher risk of prematurity, approximately 7% can have more than six times greater risk, and 3% of global population may have around 10 times higher risk of having a preterm birth. Therefore, we suggest that if the health system allows the possibility to perform a screening for preterm birth using transvaginal ultrasound to identify short cervix in the same ultrasound appointment as the second trimester morphological ultrasound (between 20–24 weeks gestation) universal screening should be recommended. This opportunity may reduce costs associated to the preterm birth screening implementation.

Another limitation of this systematic review is the significant heterogeneity between the studies in numerous aspects. The first question is regarding the origin of the studies: we had one third of the studies and the study population from the United States. However, when conducting a large-scale systematic review, it is natural to suppose that certain countries with more scientific investment present more studies with the necessary criteria to be included in the systematic review.

Also, It was not possible to assemble all the subjects into one large cohort because the studies did not prioritize the same features to report. We opted to extract as much information as provided by the original paper concerning cervical length and relate it, where possible, to the outcome and to obstetric, demographic and anthropometric variables. Therefore, our greatest difficulty in this review was extracting homogeneous information. Compiling data is a

common problem in the medical literature. If we consider completed studies, one possible method to aggregate data is by individual patient data (IPD) meta-analysis, a method where all raw information is combined and analyzed as if it was all part of one large trial/cohort. This could be the next approach following this review.

Regarding studies that are yet to begin, an interesting proposal comes from the Core Outcomes in Women's and Newborn Health (CROWN) Initiative, a journal consortium that intends to establish sets of outcomes according to specific conditions. Predefining common results before the stage of study design would allow better agreement on interpreting the information altogether [45].

One strength of our review is the overall good quality of included papers, especially considering that the texts were evaluated based on all aspects of the original research, not only focusing on our main variable. One other important strength is that the gestational age of 16 to 24 weeks, chosen to perform this review, is aligned with experts' recommendations [19] and relies on the fact that before 16 weeks there is no significant modification in cervical length associated with preterm birth [46], and after 24 weeks, due to threshold limits of fetal viability, there are many other confounding factors associated with corticotherapy and tocolysis treatment [47]. Clinical practice supports studies during this period as this gestational age range is crucial for the implementation of different approaches for preventing preterm birth if a risk factor is identified.

Even with the vast number of publications and data we were able to put together in the current review, including a total of 363,431 women, significant losses in the revision process were seen in the group of articles excluded because they exceeded the gestational age. Moreover, another expressive loss was the unquantifiable number of women undergoing universal screening in large clinical trials. Both difficulties could probably also be resolved by using an IPD meta-analysis.

## Conclusions

Cervical length ranges vary across populations and different income countries. This should be considered and interventions should be offered cautiously, especially in borderline cases. It is possible that one only cutoff to define the short cervix diagnosis cannot identify correctly this risk factor for preterm birth in different populations. New studies must be considered to identify different specific curves and cutoffs for cervical length measurement in different populations.

## Supporting information

**S1 Appendix. Prisma checklist.**
(DOCX)

**S2 Appendix. Database research: Search syntaxes.**
(DOCX)

**S3 Appendix. Women classification in high or low-risk for preterm birth.**
(DOCX)

**S4 Appendix. Quality assessment tool for observational cohort and cross-sectional studies —National Heart, Lung and Blood Institute.**
(DOCX)

**S5 Appendix. Risk of bias in included articles.**
(DOCX)

**S1 Fig. Meta-analysis without previous conization.** Forest plot for cervical length measurement in patients without previous conization.
(DOCX)

**S2 Fig. Nulliparous versus multiparous.** Difference of mean of cervical length between nulliparous versus parous women.
(DOCX)

**S3 Fig. Adolescents versus adults.** Comparison of cervical length measurements in adolescents and adults.
(DOCX)

**S1 Table. Excluded articles classified according to exclusion reasons.**
(DOCX)

**S2 Table. Main characteristics of included articles.**
(DOCX)

**S1 Data. Complete dataset.**
(XLSX)

## Author Contributions

**Conceptualization:** A. Borovac-Pinheiro, M. S. França, R. Passini, Jr., B. W. Mol, J. G. Cecatti, R. C. Pacagnella.

**Data curation:** R. C. Pacagnella.

**Formal analysis:** T. G. Bortoletto, T. V. Silva, A. Borovac-Pinheiro, A. R. Hatanaka, J. P. Argenton, R. C. Pacagnella.

**Funding acquisition:** R. C. Pacagnella.

**Investigation:** T. G. Bortoletto, T. V. Silva, A. Borovac-Pinheiro, C. M. Pereira, A. D. Silva, M. S. França, A. R. Hatanaka.

**Methodology:** T. G. Bortoletto, T. V. Silva, A. Borovac-Pinheiro, C. M. Pereira, A. D. Silva, M. S. França, A. R. Hatanaka, J. P. Argenton, R. Passini, Jr., B. W. Mol, J. G. Cecatti, R. C. Pacagnella.

**Project administration:** T. V. Silva.

**Supervision:** R. C. Pacagnella.

**Validation:** B. W. Mol, J. G. Cecatti.

**Writing – original draft:** T. G. Bortoletto, T. V. Silva, A. Borovac-Pinheiro, C. M. Pereira, M. S. França, R. Passini, Jr., B. W. Mol, R. C. Pacagnella.

**Writing – review & editing:** T. G. Bortoletto, T. V. Silva, A. Borovac-Pinheiro, C. M. Pereira, A. D. Silva, M. S. França, A. R. Hatanaka, J. P. Argenton, R. Passini, Jr., B. W. Mol, J. G. Cecatti, R. C. Pacagnella.

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
