## [Decision Letter · Decision Letter 0]

1 Dec 2020

PONE-D-20-34675

Cervical length varies considering different populations and gestational outcomes: results from a systematic review and meta-analysis

PLOS ONE

Dear Dr. Pacagnella,

Thank you for submitting your manuscript to PLOS ONE. After careful consideration, we feel that it has merit but does not fully meet PLOS ONE’s publication criteria as it currently stands. Therefore, we invite you to submit a revised version of the manuscript that addresses the points raised during the review process.

Three experts in the field handled your manuscript, and we are very thankful for their time and contributions. Although interest was found in your study, several comments arose that must be addressed. Notably, there is a request for additional data analyses; the apparent bias of the data selection needs to be acknowledged; and a copyeditor must be hired by the authors to proof the manuscript before resubmission.

We look forward to receiving your revised manuscript.

Kind regards,

Frank T. Spradley

Academic Editor

PLOS ONE

"he funders had no role in study design, data collection and analysis, decision to publish, or preparation of the manuscript"

3. We note that you have included the phrase “data not shown” in your manuscript. Unfortunately, this does not meet our data sharing requirements. PLOS does not permit references to inaccessible data. We require that authors provide all relevant data within the paper, Supporting Information files, or in an acceptable, public repository. Please add a citation to support this phrase or upload the data that corresponds with these findings to a stable repository (such as Figshare or Dryad) and provide and URLs, DOIs, or accession numbers that may be used to access these data. Or, if the data are not a core part of the research being presented in your study, we ask that you remove the phrase that refers to these data.Reviewers' comments:

Reviewer's Responses to Questions

**Comments to the Author**

1. Is the manuscript technically sound, and do the data support the conclusions?

Reviewer #1: Yes

Reviewer #2: Yes

Reviewer #3: Yes

2. Has the statistical analysis been performed appropriately and rigorously? 

Reviewer #1: I Don't Know

Reviewer #2: I Don't Know

Reviewer #3: Yes

3. Have the authors made all data underlying the findings in their manuscript fully available?

Reviewer #1: Yes

Reviewer #2: Yes

Reviewer #3: Yes

4. Is the manuscript presented in an intelligible fashion and written in standard English?

Reviewer #1: Yes

Reviewer #2: Yes

Reviewer #3: Yes

5. Review Comments to the Author

Reviewer #1: The authors performed a comprehensive systematic review of manuscripts that measured cervical length by transvaginal ultrasound among pregnant women at 16-24 weeks gestation from different continents. They also assessed associations between cervical length and body mass index (BMI), age and pregnancy outcome. The study is of considerable academic interest and is well written and readily understandable. Their findings confirmed numerous prior studies of associations between a lower cervical length and both preterm birth and low BMI. The unique contribution of their analysis is the reporting of average cervical length in pregnant women from different continents. Mean cervical length was 39.81mm, 39.37mm, 36.59mm, 35.98mm and 32.77mm in women from Europe, North America, South America, Asia and Africa.

It would have been interesting and perhaps more informative to have had a breakdown of cervical length by country in addition to continent. I imagine that this data is available for all of the included studies. Continents are huge and populations in different regions and countries can vary greatly in race, diet, vaginal microbiome composition, standard of living, endogenous diseases, rates of sexually transmitted diseases, etc., all factors that can influence cervical length. This makes the analysis of cervical length only on a continent basis of somewhat limited value. When having an initial consultation with a pregnant women knowledge of the continent where she resides will not be very useful in predicting what should be her expected cervical length.

The authors should also have available data on race and, likely, also ethnicity. It would be useful to know the racial and ethnic breakdown of cervical length in pregnant women in each continent.

A concentration of studies in selected countries also introduces a bias in the analysis of data on a continent-wide scale.

In the Abstract and throughout the manuscript it would be more accurate if the authors substitute “continent” for “countries”.

Reviewer #2: I appreciate the opportunity to review this work as the subject is definitely important and valuable. I believe that that manuscript still requires editing through a native English speaker as many sentences could have been more concisely written. shouldn't the abstract contain 'results' as heading? I think that it is important to state the criteria that were being analysed (BMI, socio-economical status, ethnicity...etc) under the methods to make it clear to the reader what is to come. Under Results, what are the other sources for the 23 articles?

Reviewer #3: In this study Authors performed a meta-analysis and systematic review on the changes of cervical length in mid trimester according to different population The subject is of interest, the analysis well performed and the response to the previous revision well done. I suggest to take into account in the discussion the recent paper from Gudicha et al AJOG 2020 doi.org/10.1016/j.ajog.2020.09.002

6. PLOS authors have the option to publish the peer review history of their article (what does this mean?). If published, this will include your full peer review and any attached files.

Reviewer #1: **Yes: **Steven S. Witkin

Reviewer #2: No

Reviewer #3: **Yes: **Giuseppe Rizzo

---

## [Author Response · Author response to Decision Letter 0]

20 Dec 2020

Reviewers Comments to the Author and Answers 

Reviewer #1: The authors performed a comprehensive systematic review of manuscripts that measured cervical length by transvaginal ultrasound among pregnant women at 16-24 weeks gestation from different continents. They also assessed associations between cervical length and body mass index (BMI), age and pregnancy outcome. The study is of considerable academic interest and is well written and readily understandable. Their findings confirmed numerous prior studies of associations between a lower cervical length and both preterm birth and low BMI. The unique contribution of their analysis is the reporting of average cervical length in pregnant women from different continents. Mean cervical length was 39.81mm, 39.37mm, 36.59mm, 35.98mm and 32.77mm in women from Europe, North America, South America, Asia and Africa.

It would have been interesting and perhaps more informative to have had a breakdown of cervical length by country in addition to continent. I imagine that this data is available for all of the included studies. Continents are huge and populations in different regions and countries can vary greatly in race, diet, vaginal microbiome composition, standard of living, endogenous diseases, rates of sexually transmitted diseases, etc., all factors that can influence cervical length. This makes the analysis of cervical length only on a continent basis of somewhat limited value. When having an initial consultation with a pregnant women knowledge of the continent where she resides will not be very useful in predicting what should be her expected cervical length.

R.Thank you for the interesting highlighted point. Indeed, it would have been interesting to have a breakdown of cervical length by country, however, many studies have data from more than one country, and in some continents, like Africa and South America, there would have only one study per country, what would have brought an additional difficulty to the analysis. We included a sentence in the results section regarding the countries’ study population: “We included one study from South Africa, Barcelona, Belgium, Botswana, Croatia, Egypt, Finland, Greece, Hong Kong, India, Iran, Italy, Japan, Romania, South Korea, Sweden, Taiwan, two studies from Tukey and UK, three from The Netherlands and Thailand, four from China and Spain, ten from Brazil and 24 from the United States. Also, five studies included women from two or more countries.”

The choice for presenting the results by continents is, however, the most appropriate for this analysis. We agree that continents can present some variety considering women’s characteristics. However, it is also important to point out that there are large similarities between countries in the same continent and considerable differences comparing different continents, which is historically confirmed by the ethnicities and formations of ancient peoples in the population and domination process of the continent’s territories. For example, there are similar characteristics for the race, BMI, and diet regarding Asian countries and these social-demographic findings are completely different from American or European countries. As a benchmark systematic review and meta-analysis study evaluating and considering cervical length varying among populations, we believe that it is important to consider continental similarities and so we prefer to use this term throughout the manuscript. We also suggest in the manuscript that new studies can refine and identify specific country populational characteristics that can influence the cervical length, providing a more accurate cervical length distribution curve for each population. 

The authors should also have available data on race and, likely, also ethnicity. It would be useful to know the racial and ethnic breakdown of cervical length in pregnant women in each continent.

R. We shared our database where is possible to find information regarding all countries involved in this analysis. Unfortunately, few studies provided information enough to analyze social-demographic characteristics. We also suggest that new studies should focus in to identify how these characteristics can influence cervical length in different populations. 

A concentration of studies in selected countries also introduces a bias in the analysis of data on a continent-wide scale.

R. Moreover, we agree that such concentration of studies can happen in with countries, however, when conducting a large-scale systematic review and meta-analysis considering all publications that respect the study's eligibility criteria, it is natural to suppose that certain countries with more scientific investment present more studies with the necessary strength to be included in the systematic review. 

We considered this in the text as a limitation:

“The first question is regarding the origin of the studies: we had one third of the studies and the study population from the United States. However, when conducting a large-scale systematic review, it is natural to suppose that certain countries with more scientific investment present more studies with the necessary criteria to be included in the systematic review.”

In the Abstract and throughout the manuscript it would be more accurate if the authors substitute “continent” for “countries”.

R. As mentioned before, we prefer to use this term throughout the manuscript

Reviewer #2: I appreciate the opportunity to review this work as the subject is definitely important and valuable. I believe that that manuscript still requires editing through a native English speaker as many sentences could have been more concisely written. 

R.

Thank you for the comments. Our paper has been professionally proofread and we provided all changes in the manuscript to meets PLOS ONE's style requirements. We resubmitted to the proofreading company.

shouldn't the abstract contain 'results' as heading?

R. 

We changed the abstract to include the 'results' as heading

I think that it is important to state the criteria that were being analysed (BMI, socio-economical status, ethnicity...etc) under the methods to make it clear to the reader what is to come. Under Results, what are the other sources for the 23 articles?

R. 

We included this information in the methods: “We also analyzed aggregated cervical length breakdown by continents, country income, women’s BMI and pregnancy outcome (preterm or term delivery).” 

Reviewer #3: In this study Authors performed a meta-analysis and systematic review on the changes of cervical length in mid trimester according to different population The subject is of interest, the analysis well performed and the response to the previous revision well done. I suggest to take into account in the discussion the recent paper from Gudicha et al AJOG 2020 doi.org/10.1016/j.ajog.2020.09.002

R. 

Thank you for your comment. The interesting article from Gudicha et al suggests a personalized cervical length assessment (a calculator) to improve preterm birth prediction. We agree that cervical length may vary and populational characteristics have to be considered to predict preterm birth, however, our article did not focusses in to predict preterm birth. Our main idea was to demonstrate the pattern of cervical length in different populations.

---

## [Decision Letter · Decision Letter 1]

7 Jan 2021

Cervical length varies considering different populations and gestational outcomes: results from a systematic review and meta-analysis

PONE-D-20-34675R1

Dear Dr. Pacagnella,

We’re pleased to inform you that your manuscript has been judged scientifically suitable for publication and will be formally accepted for publication once it meets all outstanding technical requirements.

Kind regards,

Frank T. Spradley

Academic Editor

PLOS ONE

Reviewers' comments:

Reviewer's Responses to Questions

**Comments to the Author**

1. If the authors have adequately addressed your comments raised in a previous round of review and you feel that this manuscript is now acceptable for publication, you may indicate that here to bypass the “Comments to the Author” section, enter your conflict of interest statement in the “Confidential to Editor” section, and submit your "Accept" recommendation.

Reviewer #1: All comments have been addressed

Reviewer #3: All comments have been addressed

2. Is the manuscript technically sound, and do the data support the conclusions?

Reviewer #1: Yes

Reviewer #3: Yes

3. Has the statistical analysis been performed appropriately and rigorously? 

Reviewer #1: Yes

Reviewer #3: Yes

4. Have the authors made all data underlying the findings in their manuscript fully available?

Reviewer #1: Yes

Reviewer #3: Yes

5. Is the manuscript presented in an intelligible fashion and written in standard English?

Reviewer #1: Yes

Reviewer #3: Yes

6. Review Comments to the Author

Reviewer #1: This manuscript is now acceptable for publication. All questions have been satisfactorily addressed .

Reviewer #3: Authors did a great job. congratulations nicely reviewed hope to see published the paper soon. high impact predicted

7. PLOS authors have the option to publish the peer review history of their article (what does this mean?). If published, this will include your full peer review and any attached files.

Reviewer #1: **Yes: **Steven S. Witkin

Reviewer #3: **Yes: **Giuseppe Rizzo

---

## [Editor Report · Acceptance letter]

4 Feb 2021

PONE-D-20-34675R1 

Cervical length varies considering different populations and gestational outcomes: results from a systematic review and meta-analysis 

Dear Dr. Pacagnella:

I'm pleased to inform you that your manuscript has been deemed suitable for publication in PLOS ONE. Congratulations! Your manuscript is now with our production department. 

Kind regards, 

on behalf of

Dr. Frank T. Spradley 

Academic Editor

PLOS ONE